# The origin of heterogeneous nanoparticle uptake by cells

Paul Rees[1,2,4], John W. Wills[3,4], M. Rowan Brown[1], Claire M. Barnes[1] & Huw D. Summers[1]

Understanding nanoparticle uptake by biological cells is fundamentally important to wide-ranging fields from nanotoxicology to drug delivery. It is now accepted that the arrival of nanoparticles at the cell is an extremely complicated process, shaped by many factors including unique nanoparticle physico-chemical characteristics, protein-particle interactions and subsequent agglomeration, diffusion and sedimentation. Sequentially, the nanoparticle internalisation process itself is also complex, and controlled by multiple aspects of a cell's state. Despite this multitude of factors, here we demonstrate that the statistical distribution of the nanoparticle dose per endosome is independent of the initial administered dose and exposure duration. Rather, it is the number of nanoparticle containing endosomes that are dependent on these initial dosing conditions. These observations explain the heterogeneity of nanoparticle delivery at the cellular level and allow the derivation of simple, yet powerful probabilistic distributions that accurately predict the nanoparticle dose delivered to individual cells across a population.

[1] Centre for Nanohealth, Swansea University College of Engineering, Fabian Way, Crymlyn Burrows, Swansea SA1 8EN, UK. [2] Broad Institute of MIT and Harvard, 415 Main Street, Cambridge, MA 02142, USA. [3] Biominerals Research, Cambridge University Department of Veterinary Medicine, School of Biological Sciences, Madingley Road, Cambridge CB3 0ES, UK. [4] These authors contributed equally: Paul Rees, John W. Wills. Correspondence and requests for materials should be addressed to P.R. (email: p.rees@swansea.ac.uk) or to J.W.W. (email: jw2020@cam.ac.uk)

There is a recognised demand for quantifiable and robust metrics for assessing the uptake of nanoparticles by cells driven by fields of research from nanosafety assessment to drug delivery[1–3]. In a typical in vitro study, a specific dose of nanoparticles is administered into a biological medium containing cells. The subsequent delivery of the nanoparticles to the cells is known to be a highly complicated process involving diffusion and/or sedimentation[4] of the nanoparticles in the liquid medium[4,5]. Nanoparticle agglomeration is further dependent on the content of the media[6] and the functional coating on the particles, and can also dramatically alter cellular uptake dynamics[6–8]. Upon arrival at the cell surface, there are multiple processes by which nanoparticles can be internalised by the cell, e.g., phagocytosis, endocytosis, direct trans-membrane transport, etc., and once internalised within the cell, there are many locations to which nanoparticles are routed[7]. This multitude of factors has led to much debate around the appropriate physico-chemical parameters to consider when quantifying nanoparticle cellular delivery, e.g., size, shape, surface area and charge, etc.[8–10].

Despite the vast number of factors that affect the cellular uptake of nanoparticles, we have previously shown that because the number of particles internalised by cells is random, the dose acquired by each cell can be described by well-defined probability distribution functions[11]. Given the number of nanoparticles in the medium for a typical exposure, and the size of particles relative to the cell, it is appropriate to assume that the delivery to the cell surface corresponds to a Poisson arrival process, i.e., a series of independent random events in time. However, measured distributions of nanoparticle-loaded vesicles (NLVs) deviate from the simple Poisson case and exhibit over-dispersion[11]. This indicates the presence of additional processes, beyond just particle arrival at the membrane, which play a role in determining particle uptake. It also raises the question of whether the prediction of the cellularly internalised dose based on mechanism-related models is possible—given the complexities suggested by the over-dispersed characteristics.

To get to the root of this heterogeneity, here we use high-throughput microscopy and image analysis to study nanoparticle delivery for a range of dosing conditions and cell lines. This allows us to measure the number of nanoparticles encapsulated in individual vesicles (in excess of $10^5$ per exposure condition) across tens of thousands of cells, yielding highly accurate probability distributions of acquired dose at the vesicle and whole-cell level. Our approach reveals the fundamental processes that inevitably lead to the random nature of nanoparticle dosing, and provides a simple predictive equation that allows researchers to determine the cell-to-cell variation in nanoparticle uptake that can be expected for their particular cell line, nanoparticle and dose conditions.

## Results

**High-throughput microscopy strategy and NLV over-dispersity**. In this paper, we measure the uptake of commercially available, ~8 nm Qtracker 705 quantum dot nanoparticles (physico-chemical characteristics described in Supplementary Table 1) by endocytosis for A549 (adenocarcinomic human alveolar epithelial cells) and BEAS-2B (normal human bronchial epithelial cells) exposed to 0.5, 2, 4 or 5 nanomolar administered doses for durations of 0.5, 1 or 2 h. For each experiment, we used CellProfiler[12] to segment the nucleus, cell membrane and any NLV within each cell object (Fig. 1a,b and Supplementary Figs. 1–4). The CellProfiler pipeline (provided alongside sample data, Supplementary Note 1 and Supplementary Data 1) enabled precise measurement of each cell's area (Fig. 1c), the number of NLVs per cell (Fig. 1d), as well as the fluorescence intensity of every NLV across 105 fields-of-view collected per experiment. In

previous work correlating fluorescence measurements against counts of nanoparticle numbers per vesicle using electron microscopy, we verified Qtracker fluorescence as an excellent surrogate measure of the total number of nanoparticles per vesicle[13]. Use of a fluorescent nuclear marker further allowed us to measure the DNA content of each cell, and to relate this to the functional state of the cell in terms of cell cycle position (Fig. 1e). The large number of cells (>10,000) and endosomes (~0–40 per cell) analysed per experiment provided statistically relevant numbers enabling us to generate accurate probability density functions to describe cell area, NLVs per cell and nanoparticle loading per vesicle across each cell population (exact cell/NLV $n$ sizes for all 12 exposures are provided, Supplementary Figs. 3, 4). The probability distribution describing the number of NLVs per cell for each combination of nanoparticle dose and exposure time was over-dispersed, i.e., the variance is greater than the mean, confirming previous studies[11,13] (Fig. 1d).

**Dose per cell versus dose per endosome**. Considering the results, the mean number of NLVs per cell increases linearly with increasing administered dose and duration of exposure as expected (Fig. 2a–d). However, somewhat surprisingly, the fluorescence intensity distributions of the NLVs (equating to the number of nanoparticles encapsulated within the vesicle) are independent of these experimental conditions (Fig. 2e–g, further results shown Supplementary Fig. 5). This indicates that the distribution of the nanoparticle dose encapsulated in each vesicle is highly similar for both cell lines and is fixed, being independent of the administered dose and exposure duration over a 16-fold variation in the dose-time product. Instead, the higher delivered cellular dose that follows increasing exposure manifests from an increase in the number of NLVs, and not from the loading of greater numbers of nanoparticles into individual endosomes. This implies that the endosomal loading is primarily determined by endocytosis dynamics rather than the particle arrival kinetics under these dosing conditions.

**A probabilistic model to describe nanoparticle uptake**. Given these observations, we can derive a complete analytical model that describes the statistical distribution of NLVs (and therefore also the distribution of the nanoparticle dose captured by each cell) for a given nanoparticle administered dose and exposure duration. The rate of formation of endosomes by a cell can be described as a spatiotemporal Poisson process, which is dependent on the area and turnover rate of the cell membrane[14]. The distribution of cell area across a population can be approximated by a gamma function (Fig. 1c and Supplementary Figs. 6, 7), which means the endosome formation rate per cell is then given by a negative binominal distribution (see Methods for derivation). We note that this assumes that the cell area is constant during nanoparticle exposure, which might become less valid over long exposure times or for high doses of nanoparticles where toxicity perturbations can lead to cell shrinkage[15]. The generation of an NLV depends on the arrival of at least one nanoparticle/agglomerate at the site of a nascent endosome. The particle interaction also has to take place within the lifetime of endosome formation at the membrane, which is typically in the range of 40–400 s[16]. The resulting vesicular dose is therefore dependent upon the spatial and temporal dynamics of both forming endosomes and particle-binding events (Fig. 3a). This interplay of processes determines whether particles arrive into the same endosome and thus increase the intra-vesicle dose, or bind to new vesicles to increase the NLV count. In practice, for typical nanomolar dose ranges, the endosome formation time is short compared to the inter-event time for particle arrival, and so the

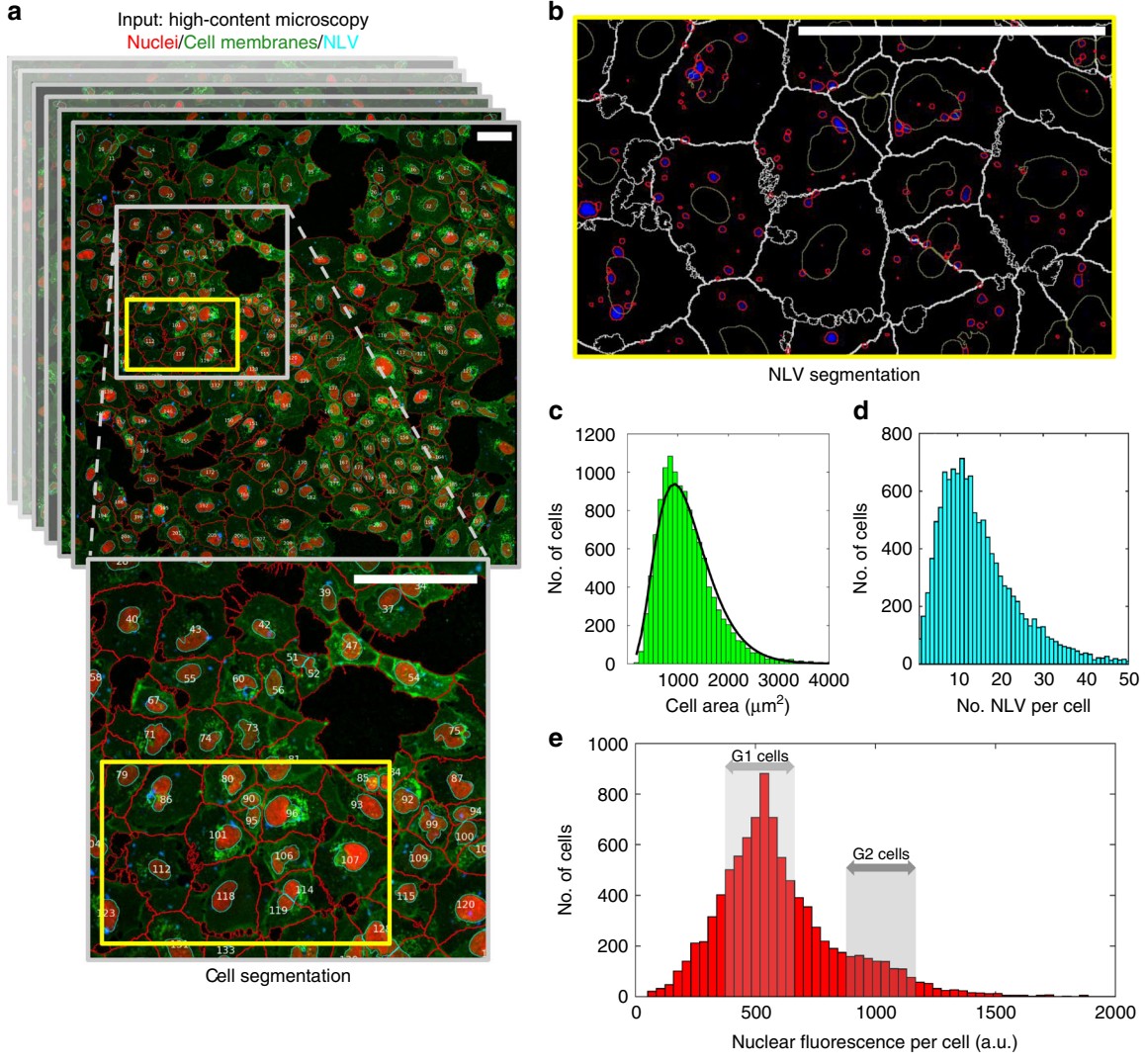

**Fig. 1** Image-based analysis of nanoparticle delivery to adherent cells. **a** A typical field-of-view (taken from >100 per experiment) imaged by laser scanning confocal microscopy of lung adenocarcinoma A549 cells exposed to a 2.0-nM dose of Qtracker® 705 quantum dot nanoparticles for 1 h. Cell identification numbers alongside nuclear and cell membrane segmentation masks achieved by image analysis (see Methods) are shown as blue and red lines, respectively. **b** For each cell (segmentation outlines shown), individual nanoparticle-loaded vesicles (NLVs) were also segmented (red outlines). **c–e** In this way, image analysis allowed nuclear, cell and NLV features (e.g., size, shape and fluorescence intensity) to be measured for ~$10^4$ cells and ~$10^5$ NLVs for each exposure condition (i.e., dose–time combination). This allowed factors such as area (**c**), number of NLVs (**d**) and the DNA content (**e**) of each cell to be measured, and allowed probabilistic models to be constructed for statistically defensible cell populations (e.g., a gamma function to describe cell area distributions, black line in **c**). (Scale bars = 100 μm.) The underlying data are provided in the BioStudies database under the accession code S-BSST249 and in Supplementary Data 1

overwhelming probability is that particles bind to newly forming endosomes. In a typical cell, ≫$10^2$ endosomes form within a 1-h period[14] (Supplementary Fig. 8), and so the observation of ~1–40 NLVs per cell (Fig. 3b) confirms this hypothesis.

The observation that the arrival of a nanoparticle/agglomerate into a forming endosome is slow enough that the endosome captures at most only one event allows us to thin the coupled endosome formation–particle capture Poisson processes[17] and arrive at a negative binomial probability distribution describing the number of NLVs per cell, $N$ due to exposure of cell population to nanoparticle concentration $C$ for a time, $t_:$

$$p(N, r, p) = \frac{\Gamma(r + N)}{N!\Gamma(r)} p^N (1 - p)^r, \tag{1}$$

where $r = \alpha$, $p = \beta\lambda Ct/(1 + \beta\lambda Ct)$ and $\alpha$ and $\beta$ describe the gamma distributions of cell area (see Methods for derivation and

assumptions). The shape factor $\alpha$ and scale factor $\beta$ are determined by fitting the measured cell area distributions (Supplementary Figs. 6, 7), $\lambda$ is the formation rate of NLVs in terms of concentration per unit time, per unit area. We are therefore able to predict the number of NLVs using a probabilistic model with just one scaling parameter, $\lambda$, which remains constant across all exposure doses and times.

The mean number of NLVs per cell described by the negative binomial distribution in Eq. 1 can be derived to be $\nu_{nb} = \lambda \times \alpha\beta \times Ct$ (see Methods for derivation), therefore the least squares fit to the mean NLV versus dose–time product (Fig. 2c, d) can be used to derive a value of $\lambda$ of 0.00107 nM$^{-1}$ h$^{-1}$ m$^{-2}$ for BEAS-2B and 0.00135 nM$^{-1}$ h$^{-1}$ m$^{-2}$ for A549. Using these values, there is an excellent agreement between the experimental and theoretical distributions for all three doses shown and for an increased exposure time of 2 h (Fig. 3b and Supplementary

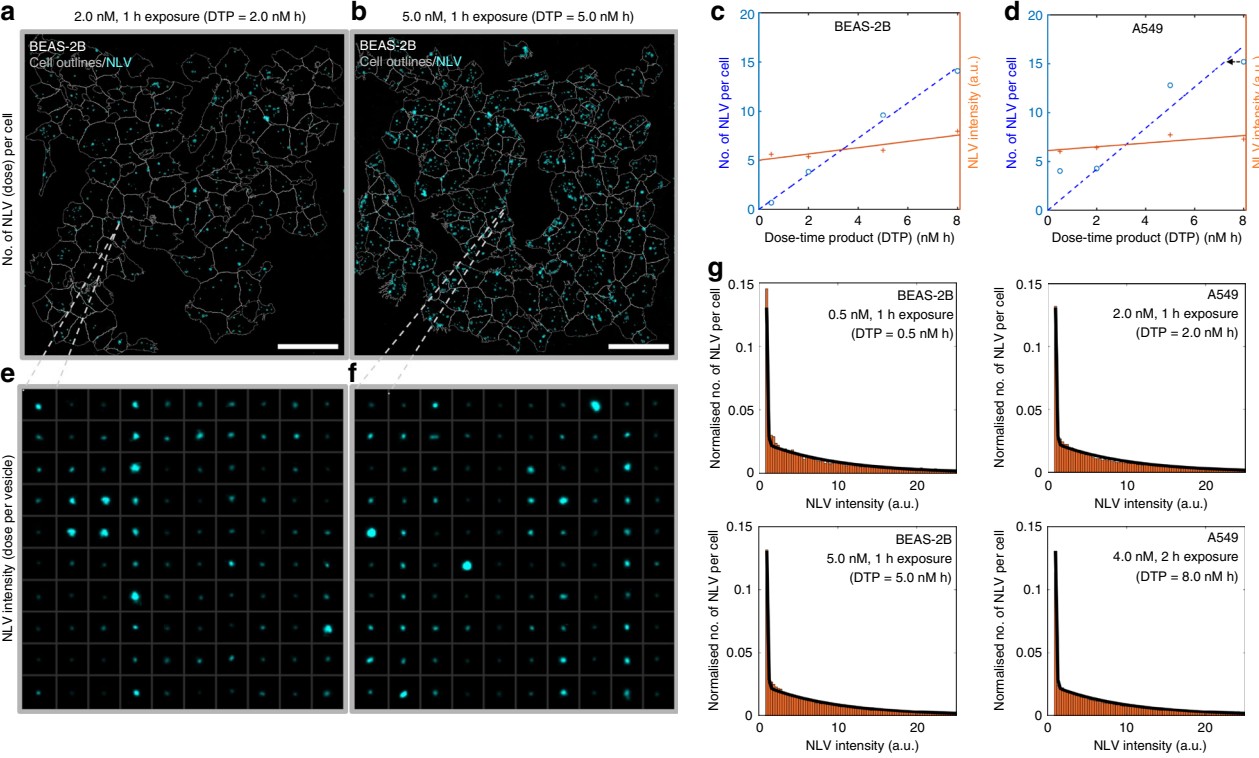

**Fig. 2** NLV analysis across different cell lines and exposure conditions. **a**, **b** Typical fields-of-view for the BEAS-2B cells demonstrating the expected increase in the number of NLVs per cell that occurs when the nanoparticle exposure dose-time product (DTP) was increased from 2 nM h (**a**) to 5 nM h (**b**). **c**, **d** Taken across all image data, the mean number of NLVs per cell (blue circles) and the mean intensity of each NLV (orange crosses) are plotted versus the dose–time product for the BEAS-2B and A549 cells. The blue and orange lines represent least squares first-order polynomial fits to the mean number of NLVs per cell or NLV intensity versus the dose–time product (respectively). The plots show that whilst the mean number of NLVs per cell (blue dashed line) increases with increasing exposure, the NLV intensity (i.e., proportional to the number of nanoparticles per vesicle) (orange line) remains remarkably similar. **e**, **f** This stability in the nanoparticle dose per endosome is visually apparent when 100 NLVs are randomly chosen from the two exposures shown in **a/b**: the montaged panels of NLV look near identical despite a 2.5-fold increase in the dose–time product. **g** Four typical NLV intensity histograms chosen from both cell lines and across the full 16-fold range of the dose–time product. The black line is the same, simultaneous fit to all of the histograms, demonstrating the insensitivity of the nanoparticle dose per vesicle to the initial administered dose or exposure duration. (Scale bars = 100 μm.) The underlying data are provided in the BioStudies database under the accession code S-BSST249

Fig. 9). The ab initio prediction of a negative binomial distribution is worthy of note as this probability function was found to be the best curve-fit model of the measured NLV distribution reported in previous work[11]. The predicted total nanoparticle fluorescence per cell (Eq. 13) using these distributions is also in excellent agreement with the experiment (Fig. 3c and Supplementary Fig. 9) (N.B., Supplementary Note 2 contains a full breakdown of all model parameters and instructions for fitting. Supplementary Data 2 additionally contains an Excel version of the model, which can be used to apply the model using results obtained from any image analysis routine, and without dedicated programming experience).

**The role of cell cycle position on nanoparticle uptake**. This prediction can be used to explore the effect of cell state on the uptake of nanoparticles, and we demonstrate this capability by clarifying the role of cell cycle on nanoparticle uptake—a subject of some current controversy[18–20]. The exposure times in the experiments herein were specifically chosen to isolate and hence allow investigation of the mechanisms of particle uptake. The short duration exposure ensures that the effect of vesicle fusion is minimised[21] and that the depletion of available nanoparticles from the exposure solution is limited enough not to impact the results[22]. The short experiment timeframe also ensures that the uptake occurs in a limited window within a much longer cell cycle (~1 cf. 24 h) and so,

cells may be viewed as being in a quasi-stationary state during the exposure, i.e., there is minimal progression through cell cycle during the particle uptake period. The cell cycle positions of individuals within the populations used in the experiment are asynchronous and therefore we would expect the population sample to contain cells from all points across the cell cycle. However, we note that the same parameter value for $\lambda$, the endosome formation rate, is used for each cell type irrespective of the dosing conditions. This implies that the rate of uptake is independent of cell cycle. To further investigate this, we gated on regions of the nuclear fluorescence distribution to select G1 and G2 cells based on the DNA content of their nuclei (Fig. 4a). The cells in G2 clearly show increased numbers of NLVs compared with those in G1 (Fig. 4b); however, given the nature of the cell cycle, the cells in G2 are also larger than those in G1. Our model includes the area of the cell in the endosome generation distribution and therefore a more appropriate comparison is the NLV per unit area (Fig. 4c). When corrected for cell area, the dose distributions for G1 and G2 cells are identical (null hypothesis accepted $p$ value = 0.586 and two-distribution Kolmogorov–Smirnov stat = 0.028 at a 0.1% significance level). These results help clarify a debate presented in recent correspondence about the role of cell cycle in controlling nanoparticle dose[18,20] as the ambiguity introduced by confounding factors of different cell area and time integration of dose are removed in this study. Crucially $\lambda$, the instantaneous formation rate of nanoparticle-loaded endosomes, in terms of concentration per unit area is

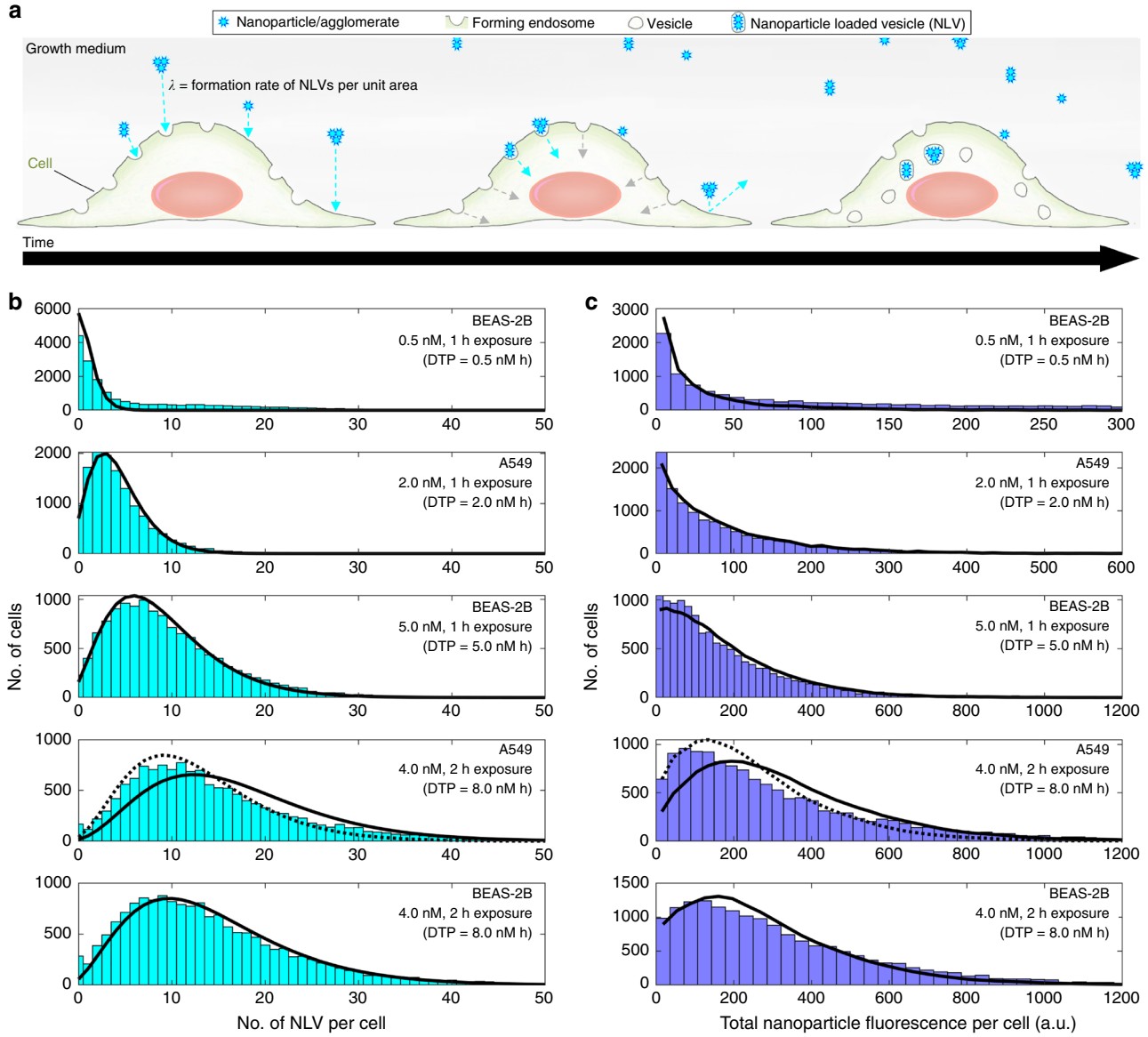

**Fig. 3** Measurements and theoretical predictions of nanoparticle cellular delivery. **a** Schematic of nanoparticle uptake by endocytosis. Nanoparticle uptake is dependent on nanoparticles and/or agglomerates arriving at the site of a forming endosome. Given the timescales of particle arrival in the context of cell membrane turnover by endocytosis, it is highly improbable that a forming endosome will encounter more than one arrival event under physiologically relevant dosing conditions. In turn, the formation of more unloaded than nanoparticle-loaded vesicles (NLVs) can be expected. **b** Histograms showing the number of NLVs formed per cell across the cell population for both BEAS-2B and A549 cells under different exposure conditions (DTP = dose–time product). The black curves are the predicted NLV distributions (see Methods, Eq. 12) using the value of $\lambda = 0.00107\ \mathrm{nM^{-1}\,h^{-1}\,m^{-2}}$ for BEAS-2B cells and $\lambda = 0.00135\ \mathrm{nM^{-1}\,h^{-1}\,m^{-2}}$ for A549 cells. We note that the fit to the 8.0 nM h data for the A549 cells is not as convincing as for the other experiments. However, the mean NLV count was lower than expected for this dose/exposure combination (see Fig. 2d), indicative of a slight error in dosing or exposure duration. If this is corrected for by using an expected value of 7.2 nM h provided by the straight line fit to the data (indicated by arrow, Fig. 2d), the distribution shown by the dotted black line is obtained. **c** Histograms of total nanoparticle fluorescence per cell across the cell population for the same exposure conditions as in **b**. The black curves are predicted cell intensity distributions (see Methods, Eq. 13). The underlying data are provided in the BioStudies database under the accession code S-BSST249

constant; therefore, the internal biological processes of the cell cycle do not impact endosome formation at the membrane.

## Discussion

In summary, we present an analytical, probabilistic model of nanoparticle dose, based on a mechanistic framework that considers the arrival of particles at the cell membrane and the availability of a forming endosome to take receipt of this delivery. In this way, the data suggests forming endosomes sample nanoparticles/agglomerates as they arrive at the cell surface on the basis of availability. Here, the model was shown to accurately predict the nanoparticle dose delivered to cells across a population over a range of exposure conditions, thus providing the ability to model pharmacokinetics. At the current time, however, the model presented was developed and tested on the basis of short exposure durations (0.5–2.0 h) rather than on the more lengthy time-integrated exposures that are typical in the field. Whilst this was extremely helpful in showing that cell area, which is correlated with the cell cycle position, is the driver of any perceived cell-cycle

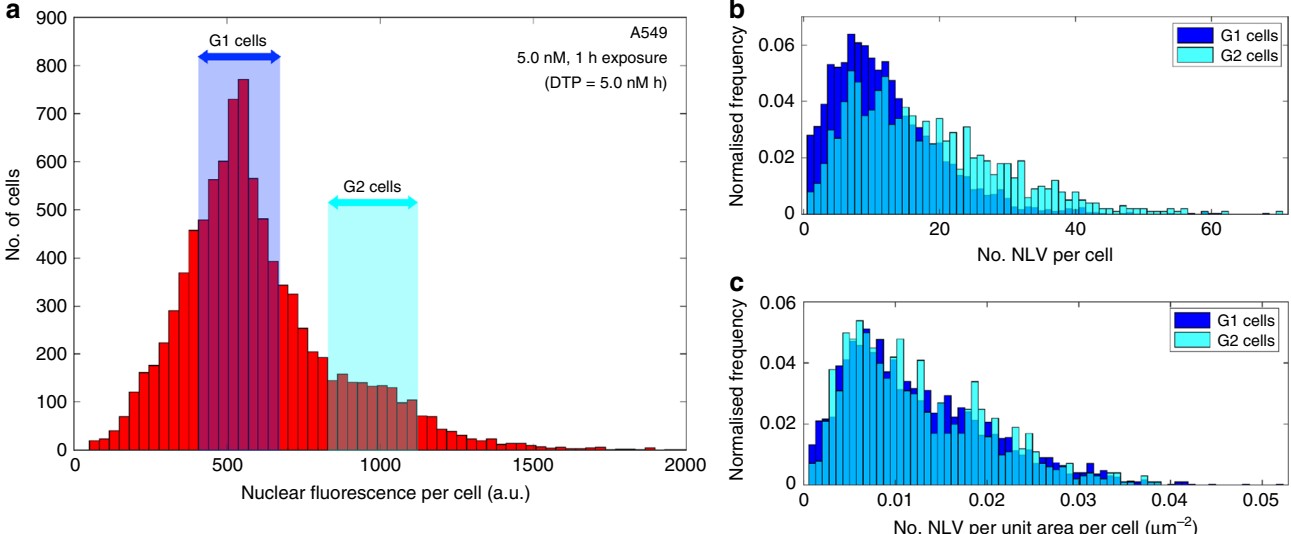

**Fig. 4** The role of cell cycle position on nanoparticle uptake. **a** A typical per-cell nuclear intensity (Hoechst 33342) histogram showing gating of cells within the characteristic G1 or G2 cell cycle positions. **b** Histograms showing the distributions of the number of NLVs per cell for the cell populations selected by the G1 or G2 cell cycle gates shown in **a**. An increase in the average number of NLVs per cell is apparent for the G2 cell population, and a two-sample Kolmogorov–Smirnov (KS) test rejects the null hypothesis at the 5% significance level that the two distributions are the same ($p = 7.447 \times 10^{-35}$, two-sample Kolmogorov–Smirnov stat = 0.229). **c** Replotted histograms of the data shown in **b**, with the number of NLVs per cell normalised against each cell's area. The two distributions now appear identical, and the null hypothesis that the two distributions are the same is not rejected even at the 0.1% significance level ($p = 0.586$, two-sample Kolmogorov–Smirnov stat = 0.028). Rather than cell-cycle-dependent uptake rates, cells at G2 present a larger area with which to capture arriving nanoparticles. The underlying data are provided in the BioStudies database under the accession code S-BSST249

dependence of nanoparticle uptake[18,20], future development should consider further adaption to account for the dilution of the nanoparticle dose that occurs through endosome inheritance upon cell division and daughter–cell formation[13]. A further consideration for effective use of the model for nanosafety applications is the effect that cell shrinkage due to toxicity[15] will play over longer exposure times.

The results presented here show that for nanoparticles, the specific route to cellular internalisation, driven by endosome formation, produces a dose profile that must be understood within a dual context of dose per vesicle and dose per cell. It is important to note, therefore, that the ability of the model to encompass this duality suggests its relevance to other modes of uptake, e.g., viral/vector delivery efficiency, microparticle and exosome uptake, etc. In contrast to molecular agents, the fundamental event underpinning particle dosing is the formation of the drug carrier (endosome)—not the contact of the drug with the cell. This, together with the rarity of event occurrence when using nanomolar concentrations typical of in vitro nanoparticle assays, results in a binary outcome of fixed dose per vesicle and a linear dependence of dose per cell upon exposure parameters. The importance of these findings to nanoparticle pharmacology is therefore context dependent. If total particle dose per cell is the required measure, then this is well behaved and suitably captured within the typical dose-response assay. However, the pharmacology of nanoparticles is often determined by the vesicular load rather than total cell dose, with particle release strategies based on charge buildup (proton sponge)[23] or other particle-driven chemical reactions[24–26], which ultimately action through disruption of the vesicular membrane. Here, the number of particles per endosome is the important factor and our results indicate limited ability to control this at the dose concentrations typically used within nanomedicine and nanotoxicology studies.

## Methods

**Cell culture and nanoparticle exposures**. Normal bronchial (BEAS-2B) and lung carcinoma (A549) cells lines were purchased from ATCC® (product numbers #CRL-9609 and #CLL-185, respectively). Cultures were maintained under standard conditions (37 °C, 5% $CO_2$, 95% humidity) in glutamine-containing Dulbecco's Modified Eagle's Medium (DMEM; #D5796, Sigma-Aldrich, UK), supplemented with 10% foetal bovine serum (FBS; #10438018, Thermo-Fisher, UK) and 1% penicillin–streptomycin (#15240062, Thermo-Fisher). Prior to the initiation of experiments, cells were maintained at ≤75% confluency for at least 72 h by routine sub-culturing involving trypsinisation (#25300054, Thermo-Fisher), centrifugation (185 × $g$, 5 min) and re-seeding in T75 culture flasks (#658175, Greiner Bio-One, UK). For imaging experiments, cells were seeded at 9000 cells cm$^{-2}$ in 4-well-chambered coverglass slides (#155383, Thermo-Fisher, UK) and returned to the incubator to adhere and acclimatise for 12 h. The cells were then pulse-loaded with commercially available Qtracker 705 quantum dot (705 nm emission, CdTe–ZnS core-shell composition) nanoparticles (#Q25061MP, Thermo-Fisher) for 0.5–2 h at concentrations of 0.5–5 nM. Non-internalised nanoparticles were then thoroughly washed out using three changes of Hank's buffered salt solution (HBSS; #14170112, Thermo-Fisher). The cells were then fixed (5 min, 4% paraformaldehyde) and cytoplasmic RNA was digested using 100 µg mL$^{-1}$ ribonuclease (#R4875, Sigma).

**Stains**. Cell membranes and nuclei were fluorescently counterstained by 10 min incubation in HBSS containing 5 µg mL$^{-1}$ wheat germ agglutinin-AlexaFluor 555 conjugate and 1 µg mL$^{-1}$ Hoechst 33342 (#W32464 and #62249, respectively, Thermo-Fisher). Counterstained cells were then washed with three changes of HBSS prior to mounting in 1 M Tris-buffered glycerol containing 25 mg mL$^{-1}$ diazabicyclo[2,2,2]octane (D27802, Sigma) as an antifade reagent.

**Image acquisition**. Images were collected using a LSM-710 laser scanning confocal microscope equipped with a 20×/0.75 objective (Carl-Zeiss, UK). Care was taken to use an optical section that exceeded the depth of the cells to ensure all NLV events were captured. Use of an automated stage and multi-position mode allowed >100 images to be collected semi-automatically for each dose–time exposure combination from across the culture well, and typically yielded ~$10^4$ cells or ~$10^5$ NLVs per experiment. Images were collected in 16-bit at 2048 × 2048 pixel resolution yielding a field-of-view coverage of ~710 µm$^2$ per image.

**Image analysis**. Images were visually inspected to remove out-of-focus or debris/artefact-containing fields. The freely available CellProfiler software[12] was then used to segment each cell's nucleus, membrane-delineated cell outline and all constituent NLVs. We include the complete image analysis pipeline used in

CellProfiler to segment the cells and measure the shape and integrated intensity of the fluorescent stains in each resulting object alongside sample images from the microscope in Supplementary Note 1 and Supplementary Data 1. We also include typical images of the segmentation for all dose–time exposure combinations to demonstrate the robustness of the in situ measurement approach (Supplementary Figs. 1–4). In brief, cell nuclei and NLVs were directly segmented as 'primary objects' using the Hoechst 33342 and Qtracker 705 signals, respectively. The cell bodies themselves were identified as 'secondary objects' using a modified watershed approach; using the nuclei as seeds for each cell-object before propagation to find the cell boundaries in the WGA-555 cell membrane channel. Care was taken to isolate the true NLV signal against background by thresholding the channel against an image stack treated/imaged identically but unexposed to Qtracker 705 (shown, Supplementary Figs. 3, 4). Before commencing data analysis, cells partially obscured at the edges of the imaging frame and that fell outside the 5th and 95th percentiles of cell area were discarded to remove incorrectly segmented cell objects[27].

**Estimation of numbers of unloaded vesicles (ULVs) per cell.** As above, cells were seeded at 9000 cells cm$^{-2}$ in 4-well chambered coverglass slides and returned to the incubator to acclimatise for 12 h. The cells were then washed with one change prewarmed HBSS before addition of a 7.5-μg-mL$^{-1}$ solution of WGA-Alexa Fluor 555 conjugate for 10 min to label cell membranes. Excess stain was then washed out with HBSS, before addition of complete phenol-red-free DMEM and return of the cells to the microscope's live cell incubator for 1 h under standard conditions. Internalisation of the WGA-555 labelled membrane into endosomes over the 1 h period was then measured by quantifying the formation of bright detail foci in each cell (process described in full, Supplementary Fig. 8). Images were collected at 2048 × 2048 pixel resolution using 20×/0.75 objective yielding a field-of-view coverage of ~425 μm$^2$ per image.

**A statistical probability distribution model.** Endosomes are formed by an active area of the membrane (often decorated with endosome promoting complexes such as clathrin), which invaginates the outer membrane forming an internalised vesicle or endosome[9]. The activation of the endocytic process on the cell membrane can be described as a spatial Poisson process, and therefore the probability distribution of the number of endosomes, $N_{end}$ generated by a cell of area $A$ within a given time, $t$, is given by,

$$P(N_{end}) = \frac{(\mu A t)^{N_{end}} e^{-\mu A t}}{N_{end}!}, \tag{2}$$

where $\mu$ is the endosome generation rate per unit area per unit time. The probability of a cell generating an NLV, $P(N_{NLV})$ relates to the conditional sequence of an endosome formation event, $P(N_{end})$ and the arrival of a nanoparticle/agglomerate at the specific cell membrane location during the endosome formation period, $P(N_{arr})$. i.e.,

$$P(N_{NLV}) = P(N_{end}) \times P(N_{arr}). \tag{3}$$

If we assume that all endosomes have the same area and turnover rate (remain at the surface for the same duration), then we can also assume that the capture of a nanoparticle cluster by an endosome is a Poisson process:

$$P(N_{arr}) = \frac{\Delta^{N_{arr}} e^{-\Delta}}{N_{arr}!}, \tag{4}$$

where $\Delta$ describes the rate of arrival of particles/agglomerates in the forming endosome. The assumption that endosomes have similar areas during formation appears to be valid[15] when the arriving particle aggregates are small, but this approximation may break down at high doses where large agglomerates form; or, where the dose is sufficient that more than one particle arrives during endosome formation[4].

**Probability of particle capture.** The probability of at least one nanoparticle cluster arriving (i.e., an NLV being formed) is given by:

$$P(N_{arr} \geq 1) = 1 - P(N_{arr} = 0), \tag{5}$$

$$\text{i.e. } P(N_{arr} \geq 1) = 1 - e^{-\Delta}. \tag{6}$$

$$P(N_{arr} \geq 1) = 1 - \left(1 - \Delta + \frac{\Delta^2}{2} - \frac{\Delta^3}{6} + \ldots\right), \tag{7}$$

$$\therefore P(N_{arr} \geq 1) \sim \Delta. \tag{8}$$

In this derivation, we have assumed that $\Delta$ is small so that higher order nonlinear terms do not contribute in the exponential expansion. This approximation has been proven to be valid for the physiologically relevant doses in previous work[21], where nanoparticle count per endosome was experimentally determined using TEM and predicted using a particle transport model developed by Hinderliter et al.[28]. This demonstrated that the arrival rate of the particles was slow enough (10–100 times slower than the endosome formation rate) that the

Poisson arrival probability distribution $P(N_{arr})$ is dominated by $N_{arr} = 0$. Therefore, the greatest probability is that no particles arrive at the site of an endosome during its formation and events where there are more than one particle ($N_{arr} > 1$) are rare. Furthermore, if we assume that the probability of arrival of at least one particle or agglomerate in the forming endosome is proportional to the concentration of the nanoparticle dose, $C$, then

$$P(N_{arr} \geq 1) = \sim \kappa C, \tag{9}$$

where $\kappa$ is a proportionality constant. Therefore, combining equations, we obtain

$$P(N_{NLV}) = \frac{(\mu A t)^{N_{end}} e^{-\mu A t}}{N_{end}!} \times \kappa C, \tag{10}$$

which is simply Poisson thinning[18] with a probability of $\kappa C$; simplifying we obtain,

$$P(N_{NLV}) = \frac{(\lambda A C t)^{N_{NLV}} e^{-\lambda A C t}}{N_{NLV}!}, \tag{11}$$

where $\lambda = \mu \kappa$ is the rate of uptake of at least one particle/agglomerate via endocytosis per unit concentration per unit area. We note that the analysis has assumed that the arrival of the nanoparticles at the surface has not initiated the endosome formation, which is believed realistic for the Qtracker nanoparticles we have used[29,30]. If, however, this was the case, the model would still be valid—just with $\kappa C$ tending towards 1—as, nearly every arriving particle would be expected to form an NLV.

The measurement of many cells allows us to measure the distribution of area for a cell population (Fig. 1c) and the most appropriate mathematical form to describe these data is a gamma distribution (Supplementary Figs. 6, 7), described by a shape factor $\alpha$ and scale factor $\beta$.

Assuming the endocytic rate per unit area, $\lambda$ to be constant for all cells in the population, then the term ($\lambda A C t$) is defined by the area gamma function. A Poisson process with a rate defined by a gamma function is a negative binomial[31]. Therefore, the probability of NLVs is given by:

$$p(N_{NLV}, r, p) = \frac{\Gamma(r + N_{NLV})}{N_{NLV}! \Gamma(r)} p^N (1 - p)^r, \tag{12}$$

where now $r = \alpha$ and $p = \beta \lambda C t / (1 + \beta \lambda C t)$.

To calculate the total nanoparticle fluorescence per cell, we simply convolve the NLV per cell distribution with the intensity of the NLV ($I_{NLV}$) distribution.

$$I_{cell}(N_{NLV}) = \sum_{0}^{N_{NLV}} N_{NLV} \int_{0}^{\infty} p(I_{NLV}) I_{NLV} dI_{NLV}. \tag{13}$$

The integral in Eq. 13 can be performed stochastically by sampling from the experimental histograms, or numerically by integrating the theoretical fit (Fig. 2g). We note that the assumption that the endosome formation rate is much faster than the particle arrival rate allows the derivation of the simple analytic expression for the NLV distribution. However, even if this assumption breaks down for very high dose/duration exposures, or due to vesicular fusion at longer exposure periods, a simple numerical model can be used by applying the same principles. Likewise, for longer exposure times, cell division, where the NLVs are inherited by the daughter cells (effectively diluting the number of NLVs per cell) must be considered in any predictive model[32]. Clearly this invalidates the simple analytic expression for NLV uptake and a more complicated numerical model is required.

**Mean dose per cell.** The experimentally determined dose per cell is seen to be a product of concentration and the duration of exposure, i.e., $\nu \propto C \times t$. To relate this to the statistical model presented above, we require the mean of the negative binomial PDF, $\nu_{nb}$:

$$\nu_{nb} = \frac{pr}{1 - p}. \tag{14}$$

Substituting for $p$ and $r$, using the expressions given above, results in the expression:

$$\nu_{nb} = \lambda \times \alpha \beta \times C t. \tag{15}$$

The $\alpha$, $\beta$ and $\lambda$ terms are constant, so Eq. 15 shows that the probabilistic model for NLV number also scales linearly with the time/dose product. (N.B., Supplementary Note 2 contains a full breakdown of all model parameters and instructions for fitting. Supplementary Data 2 additionally contains an Excel version of the model, which can be used to apply the model using results obtained from any image analysis routine, and without dedicated programming experience).

**Reporting summary.** Further information on research design is available in the Nature Research Reporting Summary linked to this article.

## Data availability

Example raw microscopy files and the complete CellProfiler image analysis pipeline are provided in Supplementary Data 1. All raw confocal data, as well as the image analysis measurement files used to produce the results and derive the model, alongside all code to produce each component of the Figures are provided in the BioStudies database under

the accession code S-BSST249. Supplementary Note 2 additionally provides a complete list of all model parameters alongside dedicated model fitting instructions. Supplementary Data 2 further provides an Excel version of the nanoparticle uptake model, which can be used with results obtained from any image analysis routine, and without dedicated programming experience. A reporting summary for this article is available as a Supplementary Information file. All other data supporting the findings of this study are available from the corresponding authors on reasonable request.

## Code availability

All computer code necessary to reproduce the Figures is provided in the BioStudies database under the accession code S-BSST249.

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

## Acknowledgements

J.W.W. would like to acknowledge Girton College and the Herchel Smith Fund of Cambridge for providing him with a post-doctoral Fellowship. The authors are grateful to J.J. Powell and S.H. Doak for their critical insights. This work was supported by the Engineering and Physical Sciences Research Council (EPSRC; grant number EP/ H008683/1). P.R. and H.D.S. would also like to acknowledge the support of the Bio-technology and Biological Sciences Research Council (BBSRC) under grants BB/ N005163/1 and BB/P026818/1.

## Author contributions

H.D.S., P.R and J.W.W. designed the experiments. J.W.W. carried out cell culture, nanoparticle exposures, microscopy and image analysis. P.R., H.D.S., C.M.B. and M.R.B. analysed the data. P.R., J.W.W. and H.D.S. wrote the manuscript in close collaboration with all co-authors. All authors reviewed, contributed in full and approved the final version of the manuscript.

## Additional information

**Competing interests:** The authors declare no competing interests.

