## [Peer Review File · Nature Communications]

Reviewers' comments:

Reviewer #1 (Remarks to the Author):

I congratulate the authors to this article. In fact, a theory like the one presented by the authors is clearly needed, as it allows to break down the uptake of nanoparticles to some important parameters. From the point of an experimentalist working on nanoparticle uptake I however suggest some refinements.

1) It would be useful if the authors made a short summary/list of the fit parameters they use. It is all in the text, but a short table enlisting the variables used for the fit would help.

2) line 299: One parameter is the number of endosomes per cell area A . Here one would need to limit the theory to low NP concentrations, as already at quite moderate concentrations cells start to shrink to the onset of toxicity and thus the cell area A is no longer constant but starts to decrease. See X. Ma, R. Hartmann, D. Jimenez de Aberasturi, F. Yang, S. J. H. Soenen, B. B. Manshian, J. Franz, D. Valdeperez, B. Pelaz, N. Feliu, N. Hampp, C. Riethmüller, H. Vieker, N. Freese, A. Götzhäuser, M. Simonich, R. Tanguay, X.-J. Liang, W. J. Parak, "Colloidal Gold Nanoparticles Induce Changes in Cellular and Subcellular Morphology", *ACS Nano* 11, 7807–7820 (2017). Figure 4B.

3) line 301: "If we assume that endosomes all have the same area". This assumption is actually good for "small" nanoparticles, see X. Ma, R. Hartmann, D. Jimenez de Aberasturi, F. Yang, S. J. H. Soenen, B. B. Manshian, J. Franz, D. Valdeperez, B. Pelaz, N. Feliu, N. Hampp, C. Riethmüller, H. Vieker, N. Freese, A. Götzhäuser, M. Simonich, R. Tanguay, X.-J. Liang, W. J. Parak, "Colloidal Gold Nanoparticles Induce Changes in Cellular and Subcellular Morphology", *ACS Nano* 11, 7807–7820 (2017). Figure 4C, where the area per lysosome has been measured versus the NP exposure concentration and has been found to be more or less constant. This agrees well with Figure 4D of Ma et al and the conclusion of the authors, that more NPs mean more endosomes/lysosomes, but the amount of NPs per endosome/lysosome would be more or less the same. Some TEM data in which individual NPs inside endosomes/lysosomes can be counted and thus their number determined might be a nice supplemental information (note: with fluorescent NPs one can not completely exclude fluorescence quenching, when they are squeezed inside endosomes/lysosomes, and thus fluorescence intensity may not be linear with the number of particles per endosome/lysosome). There is also a restriction to the constant size of endosomes/lysosomes. When going to micrometer sized particles there are lysosomes with one, with two, etc. particles, and in this case the size of each lysosome in fact increases. This is most likely the case when the size of one particle exceeds the size of natural endosomes, lysosomes. For an image of this see for example L. Kastl, D. Sasse, V. Wulf, R. Hartmann, J. Mircheski, C. Ranke, S. Carregal-Romero, J. A. Martínez-López, R. Fernández-Chacón, W. J. Parak, H.-P. Elsaesser, P. Rivera Gil, "Multiple Internalization Pathways of Polyelectrolyte Multilayer Capsules into Mammalian Cells", *ACS Nano* 7, 6605–6618 (2013), Figure 2B. Here one sees that one lysosome is "blown" up by having several particles inside and here the assumption that endosome/lysosome size remains constant would no longer hold true.

4) What about cell division. Some cells can divide quite rapidly, on the time scale of hours, which is not much higher than the time scale of NP uptake, which is also on the time scale of hours. The number of NPs per cell will be affected by rapid cell division and in fact this might be also one limiting factor for the amount of NPs which can be inside each cells. The authors have considered the cell cycle (Figure 4), but I guess that simply the dilution factor of NPs based on cell division plays a big round. Data which are just accepted but not published show actually that the number of NPs per cell scales anti-reciprocal with the proliferation rate of cells. This however might obviously be completely different for in vivo exposures.

5) line 36: " However, the arrival of nanoparticles at the cell membrane is a complicated process involving diffusion and sedimentation of the nanoparticles in the liquid medium" and the following lines. In fact, agglomeration can be fitted into this model: agglomeration involves "bigger" effective particles, which then lets sedimentation win over diffusion, see N. Feliu, X. Sun, R. A. Alvarez Puebla, W. J. Parak, "Quantitative Particle–Cell Interaction: Some Basic Physicochemical Pitfalls", *Langmuir* 33, 6639-6646 (2017).

I hope the mentioned thoughts are interesting to the authors. The manuscript is great as it is, but maybe a more severe discussion about the limitations of the model would benefit. I also suggest the authors to provide some templates that other groups later on can use their evaluation procedure. Fitting of uptake data with the given model would for all groups be helpful to break down a large set of experimental data to a few fit parameters and in this way studies could be made more comparable.

Wolfgang Parak

Reviewer #2 (Remarks to the Author):

Nanoparticle uptake by cells is a complicated process involving particle diffusion, sedimentation, agglomerations and interactions with cell membranes. The cell's state also influences the nanoparticle internalization. This paper measured the uptake of commercially available quantum dot nanoparticles for different cells, namely A549 and BEAS-2B, under different drug dosages and exposure time. The results showed that the NP dose per endosome is independent of the initial administered dose and exposure duration, while it is the number of endosomes containing NP that are dependent on the initial conditions. It also showed that the internal biological process of cell cycle does not impact on endosome formation at the membrane. Furthermore, probabilistic models based on the experimental statistical results were developed to predict the NP doses delivered to individual cells. The model showed excellent agreement with experimental results.

The paper is well organized and clearly written. It is easy and enjoyable to read. There are a few minor questions needed to be addressed if possible.

1. Fig.2 c and d, the author showed the mean NLV per cell and mean intensity of each NLV. Could the author provide the standard deviation for these means to further strengthen the results?
2. As the author mentioned that size, shape, surface coating and charge may influence the cell uptake, can the author provide more details on these physical/chemical properties of the NPs used in the experiment?
3. As cells have a 3D shape, it is not clear if the author had considered the case when the NLV and NPs may be out of focus during the laser scan. Is that going to affect the total number of NLVs and NP intensity in cells?

Reviewer #3 (Remarks to the Author):

I have read this paper over a number of times, and think it should be published because it brings forward the discussion of important topics. It is an interesting paper that looks at a general topic that, surprisingly, has not been investigated much, despite its fundamental role in nanoparticle interactions. It is relevant to some recent discussions of the semantics of 'intracellular concentration' but it has broader relevance in that the precise nature of the punctate character in cells has not been closely resolved previously.

If I understand correctly there are a number of implicit assumptions that could be discussed more fully. For instance, in some places particles are seen as taking advantage of the formation of vesicles, rather than themselves stimulating their initiation, via the whole machinery of membrane adaptors and while that could be true for some events, it is not clear if all are so formed? There are several other subtle features of this type that could be further elucidated, at least so that readers can fully understand the assumptions. I do not think it necessary (or possible at this point) to decide if these are all correct assumptions, but clarifying the ideas will assist ongoing scholarship.

On the question of presentation, I think it would also be useful for the less numerate biologist in the field if some of the text was simplified (better augmented) to explain somewhat better the main conclusions on that topic of 'NLV' dynamics in the simplest possible language. There are also several typos, and peculiar size of some symbols in the equations, but none that I identify as being incorrect. These are all easily corrected, and certainly the paper is well written as it is for the more prepared reader.

In summary, an interesting contribution that should be published.

**NCOMMS-18-36921-T: Rees *et al.*,
The Origin of Heterogeneous Nanoparticle Uptake by Cells**

Response to the Reviewers' Comments –

We would like to thank the reviewers for their helpful comments and support for the manuscript. We believe that the referees' comments have allowed us to significantly improve the content and comprehensibility of the manuscript and its take-home messages – for both the biological and biophysical audiences.

In addition to the reviewers' comments below, we have also reformatted the manuscript into the *Nature Communications* style.

Corresponding with the marked-up copy of the manuscript, a point-by-point response to each comment is provided below.

Reviewer 1:	
Comment	Response to the reviewer
1.) I congratulate the authors to this article. In fact, a theory like the one presented by the authors is clearly needed, as it allows to break down the uptake of nanoparticles to some important parameters. From the point of an experimentalist working on nanoparticle uptake I however suggest some refinements.	We are grateful to the referee for their support, and for taking the time to provide such useful comments that really helped us to improve the manuscript. In particular the referee's help in providing pointers to work in the literature which provides additional support or enables focused discussion around our ideas was very much appreciated.
2.) It would be useful if the authors made a short summary/list of the fit parameters they use. It is all in the text, but a short table enlisting the variables used for the fit would help.	We agree, and we have now provided an additional file (Additional File S2) of which details all of the model parameters as well as provides detailed fitting instructions for the non-expert.

3.) line 299: One parameter is the number of endosomes per cell area A. Here one would need to limit the theory to low NP concentrations, as already at quite moderate concentrations cells start to shrink to the onset of toxicity and thus the cell area A is no longer constant but starts to decrease. See X. Ma, R. Hartmann, D. Jimenez de Aberasturi, F. Yang, S. J. H. Soenen, B. B. Manshian, J. Franz, D. Valdeperez, B. Pelaz, N. Feliu, N. Hampp, C. Riethmüller, H. Vieker, N. Freese, A. Götzhäuser, M. Simonich, R. Tanguay, X.-J. Liang, W. J. Parak, "Colloidal Gold Nanoparticles Induce Changes in Cellular and Subcellular Morphology", ACS Nano 11, 7807–7820 (2017). Figure 4B.	The referee makes a completely valid point. To address this, we have added a sentence and the suggested supporting reference to the results section (Page 5, line 185) where the concept is first introduced. We have then included a further paragraph in the discussion, which speaks to limitations of the model including the consequences of this effect; specifically including instances where the model could become invalid due to toxicity and cell shrinkage (page 7, line 413 onwards.)
4.) line 301: "If we assume that endosomes all have the same area". This assumption is actually good for "small" nanoparticles, see X. Ma, R. Hartmann, D. Jimenez de Aberasturi, F. Yang, S. J. H. Soenen, B. B. Manshian, J. Franz, D. Valdeperez, B. Pelaz, N. Feliu, N. Hampp, C. Riethmüller, H. Vieker, N. Freese, A. Götzhäuser, M. Simonich, R. Tanguay, X.-J. Liang, W. J. Parak, "Colloidal Gold Nanoparticles Induce Changes in Cellular and Subcellular Morphology", ACS Nano 11, 7807–7820 (2017). Figure 4C, where the area per lysosome has been measured versus the NP exposure concentration and has been found to be more or less constant. This agrees well with Figure 4D of Ma et al and the conclusion of the authors, that more NPs mean more endosomes/lysosomes, but the amount of NPs per endosome/lysosome would be more or less the same. Some TEM data in which individual NPs inside endosomes/lysosomes can be counted and	Again, the referee makes some very important points here which have helped us to strengthen the discussion of the model and our assumptions. We are grateful to the referee to pointing us to the work showing the vesicles are of similar size, which is an assumption we use for the model – we have now referenced this work in the description of the model (Page 11, Line 544). Likewise, the work showing the breakdown of this assumption when more particles are captured is an important consideration and as such, we have referenced this work and described that this could invalidate our approach (however it must be stressed that this is only expected at very high nanoparticle concentrations) (Page 11, Line 546).

thus their number determined might be a nice supplemental information (note: with fluorescent NPs one can not completely exclude fluorescence quenching, when they are squeezed inside endosomes/lysosomes, and thus fluorescence intensity may not be linear with the number of particles per endosome/lysosome). There is also a restriction to the constant size of endosomes/lysosomes. When going to micrometer sized particles there are lysosomes with one, with two, etc. particles, and in this case the size of each lysosome in fact increases. This is most likely the case when the size of one particle exceeds the size of natural endosomes, lysosomes. For an image of this see for example L. Kastl, D. Sasse, V. Wulf, R. Hartmann, J. Mircheski, C. Ranke, S. Carregal-Romero, J. A. Martínez-López, R. Fernández-Chacón, W. J. Parak, H.-P. Elsaesser, P. Rivera Gil, "Multiple Internalization Pathways of Polyelectrolyte Multilayer Capsules into Mammalian Cells", ACS Nano 7, 6605–6618 (2013), Figure 2B. Here one sees that one lysosome is "blown" up by having several particles inside and here the assumption that endosome/lysosome size remains constant would no longer hold true.	
5.) What about cell division. Some cells can divide quite rapidly, on the time scale of hours, which is not much higher than the time scale of NP uptake, which is also on the time scale of hours. The number of NPs per cell will be affected by rapid cell division and in fact this might be also one limiting factor for the amount of NPs which can be inside each cells. The	The referee is of course entirely correct - any cell division would effectively dilute the NLV count in the daughter cells. If this is the case, an analytic solution becomes impossible and we would have to resort to a numerical model – which has been the subject of some of our previous work (see Summers et al., ACS Nano 2013:7 6129-6137). We have added a discussion of this important point in the manuscript in the

authors have considered the cell cycle (Figure 4), but I guess that simply the dilution factor of NPs based on cell division plays a big round. Data which are just accepted but not published show actually that the number of NPs per cell scales anti-reciprocal with the proliferation rate of cells. This however might obviously be completely different for in vivo exposures.	form of a short paragraph that more thoroughly discusses the limitations of the model in its current form (Page 7, Line 413).
6.) line 36: " However, the arrival of nanoparticles at the cell membrane is a complicated process involving diffusion and sedimentation of the nanoparticles in the liquid medium" and the following lines. If fact, agglomeration can be fitted into this model: agglomeration involves "bigger" effective particles, which then lets sedimentation win over diffusion, see N. Feliu, X. Sun, R. A. Alvarez Puebla, W. J. Parak, "Quantitative Particle–Cell Interaction: Some Basic Physicochemical Pitfalls", Langmuir 33, 6639-6646 (2017).	We thank the reviewer for their helpful input and have now mentioned agglomerates in the introduction and referenced the suggested work. More specifically, we have amended the sentence in the introduction to read (Page 2, line 60): The subsequent delivery of the nanoparticles to the cells is known to be a highly complicated process involving diffusion and/or sedimentation^{4, 5} of the nanoparticles in the liquid medium^{4, 5}.
7.) I hope the mentioned thought are interesting to the authors. The manuscript is great as it is, but maybe a more severe discussion about the limitations of the model would benefit. I also suggest the authors to provide some templates that other groups later one can use their evaluation procedure. Fitting of uptake data with the given model would for all groups be helpful to break down a large set of experimental data to a few fit parameters and in this way studies could be made more comparable.	We agree entirely with the reviewer that a more detailed discussion of the limitations of the model was missing from the initial submission. We have tackled this in the revised version by raising some of these limitations as the model parameters are derived and introduced throughout the manuscript (changes detailed above) and, additionally, by adding a detailed paragraph to discussion which speaks to the limitations and / or assumptions of the model and when these might break down during experimentation (Page 7, Line 413 onwards). Given the referees' comments regarding templates to aid other groups we have now provided a full description of all model parameters and the fitting procedure in a separate document (Additional File S2). Additionally, we also

provide an EXCEL spreadsheet, into which a user need only cut and paste a list of cell areas and NLV count (from whatever image analysis tool they have used). The spreadsheet will then fit the data distributions and calculate the endosome loading rate (Additional File S3). Critically, no programming expertise is required to use this approach, which should remove a barrier to access.

Reviewer 2:

8.) The paper is well organized and clearly written. It is easy and enjoyable to read. There are a few minor questions needed to be addressed if possible.

We are grateful to the referee for their support and agree that all the suggestions made significantly help describe the experiment, model and conclusions reached.

9.) Fig.2 c and d, the author showed the mean NLV per cell and mean intensity of each NLV. Could the author provide the standard deviation for these means to further strengthen the results?

The referee is correct of course - the standard deviation should normally be included in all graphs. However, our data is very different from a normal distribution as shown by Figures 3b/c and Figure 2g. The standard deviation is therefore very large and gives a false impression of reproducibility of the data and would therefore not generally be used in this particular instance. We include the plot with error bars below to demonstrate the problem.

This of course is not an issue to us as we are able to actually fit the NLV distribution and measure the NLV intensity distribution, so we fully capture the variation in these parameters.

Inset Figure - The NLV intensity shows a large variation according to the standard deviation BUT it is completely consistent between doses (**Figure 2g**) since the data is not well described by a normal distribution. So, in this case the standard deviation is not a appropriate measure of variation – rather the graphs given in figure 2g show the consistence given that we simultaneously treat all these curves to the same fit.

10.) As the author mentioned that size, shape, surface coating and charge may influence the cell uptake, can the author provide more details on these physical/chemical properties of the NPs used in the experiment?

This is an important additional table that was overlooked at initial submission. We have now added Supplementary Table S1 to the supplementary information outlining the key physico-chemical characteristics of the nanoparticles. In line with the comments above, we have also included more dedicated discussion of how the interaction between particle and the cell membrane could affect our model and its validity (Page 7, line 413).

11.) As cells have a 3D shape, it is not clear if the author had considered the case when the NLV and NPs may be out of focus during the laser scan. Is that going to affect the total number of NLVs and NP intensity in cells?	This is an excellent point. In this work, we use a 20X objective, which meant that even when sampling at 1 airy unit, the optical section depth was far greater than the depth of cells – which tended to have the flattened ‘fried egg’-type phenotype on the surface of the cell culture wells. We are there confident we captured all the NLVs in the optical section. However, we didn’t mention this in the original manuscript and are grateful to the referee for pointing out this important consideration which we now discuss in the Methods section (Page 9 Line 476).
Reviewer 3:	
12.) I have read this paper over a number of times, and think it should be published because it bring forward the discussion of important topics. It is an interesting paper that looks at a general topic that, surprisingly, has not been investigated much, despite its fundamental role in nanoparticle interactions. It is relevant to some recent discussions of the semantics of 'intracellular concentration' but it has broader relevance in that the precise nature of the punctate character in cells has not been closely resolved previously.	We are most grateful to the referee for their supportive and insightful comments.
13.) If I understand correctly there are a number of implicit assumptions that could be discussed more fully. For instance, in some places particles are seen as taking advantage of the formation of vesicles, rather than themselves stimulating their initiation, via the whole machinery of membrane adaptors and while that could be true for some events, it is not clear if all are so formed? There are several other subtle features of this type that could be further elucidated, at	The referee makes an important point about the interaction of the cell membrane and the particles and we have now added paragraphs to both the discussion (Page 7, Line 413) and model derivation sections (Page 12, Line 562) to more carefully discuss the various scenarios. In particular, we now discuss how a particle initiating endocytosis modifies the model assumptions and the outcomes that could subsequently be expected (Page 12, Line 584).

least so that readers can fully understand the assumptions. I do not think it necessary (or possible at this point) to decide if these are all correct assumptions, but clarifying the ideas will assist ongoing scholarship.	
14.) On the question of presentation, I think it would also be useful for the less numerate biologist in the field if some of the text was simplified (better augmented) to explain somewhat better the main conclusions on that topic of 'NLV' dynamics in the simplest possible language.	We agree with the reviewer that it is essential that model and key concepts are made as accessible as possible to both the biophysical and biological audiences. To address this, we have now rewritten the discussion section and simplified the description of the assumptions and development of the model to be less mathematical. We have also added two additional files to the submission. The first details all model parameters and contains instructions for the non-expert to achieve fitting (Additional File S2). Furthermore, we have also added another supplementary file that permits model fitting via an EXCEL spreadsheet - thus enable anyone to perform the fitting analysis with their own data without dedicated programming expertise (Additional File S3).
15.) There are also several typos, and peculiar size of some symbols in the equations, but none that I identify as being incorrect. These are all easily corrected, and certainly the paper is well written as it is for the more prepared reader. In summary, an interesting contribution that should be published.	The authors would like to thank the reviewer for pointing this out. We have been through the equations and corrected all symbol sizes. We have also restructured the manuscript to the Nature Communications format and worked together to double-check for typos.
Editor's Comments	
16.) Data, Code and Software Submission:  • Provide a "Source Data File". This should at minimum contain the raw data from all figures and graphs etc. This file should be labeled "Source data". 	All data and code necessary to replicate the results were provided as source data files at the NCOMMS transfer site. The location of the source data is detailed in all figure legends.

 • The location of the source data should be specifically mentioned in all relevant figure legends • Supply all custom code and a “readme.txt” describing how to utilize it. 	The source code contains a ‘readme’ file describing exactly how to run it and what software version was used etc.
17.) Editorial Policy checklist	Submitted
18.) Reporting Requirements for Biological Research checklist	Submitted
19.) Format Requirements checklist	Submitted
20.) Code and Software Submission checklist	Submitted
21). Add a Data Availability Statement	Done - as a separate section after the Methods but before the references (Page 14 Line 667).

REVIEWERS' COMMENTS:

Reviewer #1 (Remarks to the Author):

Excellent work and also careful revision. This article should be published as it is.